# SINGLE-STREAM POLICY OPTIMIZATION

**Zhongwen Xu**[*] **and Zihan Ding**[*]
Tencent

## ABSTRACT

We revisit policy-gradient optimization for Large Language Models (LLMs) from a single-stream perspective. Prevailing group-based methods like GRPO reduce variance with on-the-fly baselines but suffer from critical flaws: frequent degenerate groups erase learning signals, and synchronization barriers hinder scalability. We introduce Single-stream Policy Optimization (SPO), which eliminates these issues by design. SPO replaces per-group baselines with a persistent, KL-adaptive value tracker and normalizes advantages globally across the batch, providing a stable, low-variance learning signal for every sample. Being group-free, SPO enables higher throughput and scales effectively in long-horizon or tool-integrated settings where generation times vary. Furthermore, the persistent value tracker naturally enables an adaptive curriculum via prioritized sampling. Experiments using Qwen3-8B show that SPO converges more smoothly and attains higher accuracy than GRPO, while eliminating computation wasted on degenerate groups. Ablation studies confirm that SPO's gains stem from its principled approach to baseline estimation and advantage normalization, offering a more robust and efficient path for LLM reasoning. Across five hard math benchmarks with Qwen3-8B, SPO improves the average maj@32 by $+3.4$ percentage points (pp) over GRPO, driven by substantial absolute point gains on challenging datasets, including $+7.3$ pp on BRUMO 25, $+4.4$ pp on AIME 25, $+3.3$ pp on HMMT 25, and achieves consistent relative gain in pass@$k$ across the evaluated $k$ values. SPO's success challenges the prevailing trend of adding incidental complexity to RL algorithms, highlighting a path where fundamental principles, not architectural workarounds, drive the next wave of progress in LLM reasoning. Our code is publicly available at `https://github.com/verl-project/verl-recipe/tree/main/spo`.

## 1 INTRODUCTION

Reinforcement learning (RL) (Sutton & Barto, 2018) has become a cornerstone for advancing the reasoning capabilities of Large Language Models (LLMs), notably the Reinforcement Learning with Verifiable Reward (RLVR) paradigm (Lambert et al., 2024; DeepSeek, 2025). Methods like Group Relative Policy Optimization (GRPO) (Shao et al., 2024; DeepSeek, 2025) have achieved remarkable success by adopting a *multi-outcome* approach, generating a group of responses for each prompt to construct an on-the-fly baseline for variance reduction. While this "group-based" paradigm has pushed the state of the art, it suffers from fundamental inefficiencies. When all responses in a group share the same outcome (e.g., all correct or all incorrect), the relative advantage collapses to zero, yielding no learning signal. This degeneracy represents a fundamental waste of computation and data. To counteract this, a series of engineering heuristics like dynamic sampling (Yu et al., 2025) have been developed. These workarounds, while functional, add significant complexity and create a less principled, more convoluted optimization process.

Group-based architectural choice also imposes a critical synchronization barrier. In distributed training, the entire group must wait for its slowest member, a bottleneck that becomes particularly acute in complex agentic tasks requiring multi-turn tool use or long-horizon reasoning (Gao et al., 2025; Xu et al., 2025; Zhipu, 2025). In these settings, interaction times are highly variable (e.g., number of interaction turns, time per interaction, etc), and a single slow-running agentic trajectory can stall its entire group, severely hindering training throughput and scalability.

We advocate for returning to the classic single-stream paradigm for policy gradient optimization (Sutton & Barto, 2018), where each training sample is a single stream of prompt-response pair. This is not a mere simplification, but a deliberate re-alignment with foundational RL principles to address the aforementioned architectural flaws. To overcome the critical challenge of high gradient

---

[*]Equal contributions, corresponding to: zhongwensxu@gmail.com, dingzihan737@gmail.com

variance in this setting, we introduce Single-stream Policy Optimization (SPO). SPO replaces the noisy, on-the-fly group baseline with three synergistic components for stable and efficient learning. First, it employs a lightweight Bayesian value tracker to maintain a persistent, temporally-informed estimate of the success probability for each prompt, serving as a low-variance baseline. Second, it normalizes advantages globally across the entire batch, avoiding the instability of per-group statistics. Finally, this architecture naturally enables an adaptive curriculum via prioritized sampling, focusing computational resources on the most informative prompts.

The benefits of this principled approach are clear: SPO is inherently more scalable and eliminates the computational waste of degenerate groups. Our experiments confirm these advantages, demonstrating that SPO consistently outperforms GRPO on challenging reasoning benchmarks, improving the absolute point gains on challenging datasets, including $7.3$ percentage points (pp) on BRUMO 25, $4.4$ pp on AIME 25, $3.3$ pp on HMMT 25, and the pass@$k$ curves of SPO are above GRPO for all $k$s. The scalability benefit is particularly pronounced in agentic settings. Our simulations, designed to model these variable-time scenarios, show that SPO's group-free design can achieve a $4.35\times$ training throughput speedup by eliminating group synchronization bottlenecks. SPO thus provides a more robust foundation for modern LLM optimization, prompting a re-evaluation of essential versus incidental complexity in the field.

## 2 RELATED WORK

Group Relative Policy Optimization (GRPO) (Shao et al., 2024) addresses the computational overhead and training instability of PPO-style algorithms (Schulman et al., 2017) by eliminating the need for a separate critic network. Instead, GRPO constructs baselines on-the-fly using multiple responses generated for each prompt. Specifically, GRPO samples a *group* of multiple responses for each prompt and normalizes the rewards within this group to have zero mean and unit variance, creating relative advantages for policy updates. However, this approach can be inefficient if all responses in a group receive the same reward (e.g., all incorrect or all correct), resulting in a zero-advantage for all samples and providing no learning signal. To address this, DAPO (Yu et al., 2025) enhances GRPO with engineering treatments like dynamic sampling, which continues generating responses until non-zero advantages are achieved, ensuring meaningful gradients.

Several other works have proposed improvements to group-based methods. Zheng et al. (2025b) introduce GRESO, an online filtering algorithm that leverages reward training dynamics to predict and skip uninformative prompts *before* generation, Qu et al. (2025) introduce a Bayesian estimation of the prompt accuracy and use it to form a bandit strategy, significantly reducing rollout overhead. Liu et al. (2025c) propose "Lite PPO", which simplifies RLVR training to only advantage normalization and token-level loss aggregation.

Other group-based approaches include RLOO (Ahmadian et al., 2024), which returns to the simpler REINFORCE (Williams, 1992; Sutton & Barto, 2018) algorithm using a Leave-One-Out baseline that treats entire generations as single actions. Similarly, Hao et al. (2025) propose On-Policy RL with Optimal Baseline (OPO), which uses a length-weighted average of rewards as an optimal simplified baseline. Despite these improvements, all group-based methods share fundamental limitations. They construct baselines from concurrently generated responses rather than persistent, historical estimates, inheriting the same core architectural constraints as GRPO: synchronization overhead and increased generation costs in distributed settings.

Moving beyond group-based methods, Brantley et al. (2025) propose $A^*$-PO, a two-stage framework for single-sample efficiency. In its first stage, $A^*$-PO performs offline estimation of the *optimal* value function $V^*$, not the policy-specific $V_\pi$. The second stage then uses this pre-computed value to build *optimal* advantage estimates $A^*$. However, $A^*$-PO has key limitations: its *fixed*, offline-computed estimate does not adapt as the policy evolves, and it is constrained by KL-regularized optimization, restricting deviation from a reference policy.

## 3 BACKGROUND

Reinforcement learning (RL) algorithms have been used to align Large Language Models (LLMs) with human preferences (RLHF) and to optimize verifiable reward signals (RLVR; e.g., (Lambert et al., 2024; Shao et al., 2024)).

### 3.1 POLICY GRADIENT AND THE REINFORCE ALGORITHM

The foundational method for this optimization is the policy gradient theorem (Williams, 1992; Sutton & Barto, 2018). For LLMs, a trajectory consists of generating a single response $y$ from a prompt $x$. The objective function is the expected reward:

$$J(\theta) = \mathbb{E}_{x \sim \mathcal{D}, y \sim \pi_\theta(\cdot|x)}[R(x, y)], \tag{1}$$

where $\mathcal{D}$ is the prompt distribution and $R(x, y)$ is the reward for generating response $y$ for prompt $x$. The gradient of this objective is given by:

$$\nabla_\theta J(\theta) = \mathbb{E}_{x \sim \mathcal{D}, y \sim \pi_\theta(\cdot|x)}[R(x, y)\nabla_\theta \log \pi_\theta(y|x)]. \tag{2}$$

This formulation gives rise to the REINFORCE algorithm (Williams, 1992; Sutton & Barto, 2018), which updates the policy by taking a step in the direction of this estimated gradient. A significant drawback of REINFORCE is the high variance of its gradient estimator. The raw reward $R(x, y)$ can fluctuate widely, leading to noisy updates and unstable training.

To mitigate high variance, a baseline $b(x)$ that is conditionally independent of the action $y$ can be subtracted from the reward. This results in an unbiased gradient estimator with provably lower variance (Greensmith et al., 2004):

$$\nabla_\theta J(\theta) = \mathbb{E}_{x \sim \mathcal{D}, y \sim \pi_\theta(\cdot|x)}[(R(x, y) - b(x))\nabla_\theta \log \pi_\theta(y|x)]. \tag{3}$$

The term $A(x, y) = R(x, y) - b(x)$ is known as the advantage. The optimal baseline that minimizes variance is the true value function $V_\pi(x) = \mathbb{E}_{y \sim \pi_\theta(\cdot|x)}[R(x, y)]$, which is the expected reward for a given prompt $x$. In practice, $V_\pi(x)$ is unknown and must be estimated. The quality of this estimation is crucial for the stability and efficiency of the RL algorithm.

### 3.2 VARIANCE REDUCTION BASELINES FOR LARGE LANGUAGE MODELS

Several strategies have been developed to estimate the baseline $b(x)$ in the context of LLM training. PPO (Schulman et al., 2017) trains a parameterized critic network $v_\phi$. However, learning $v_\phi$ is notoriously unstable and resource-intensive, as $\phi$ typically matches the size of the LLM policy parameters $\theta$.

A common approach is to construct an empirical, on-the-fly baseline from multiple samples. Group Relative Policy Optimization (GRPO) (Shao et al., 2024; DeepSeek, 2025) generates a group of $G$ responses $\{y_1, \ldots, y_G\}$ for a single prompt $x$, then uses the mean rewards of the group as the baseline $b_{\text{GRPO}}$. Another popular baseline is the Leave-One-Out (RLOO). For a given sample $y_i$, the baseline is the average reward of the other $G - 1$ samples in the group, denoted as $b_{\text{RLOO}}$:

$$b_{\text{GRPO}}(x) = \frac{1}{G}\sum_j R(x, y_j), \qquad b_{\text{RLOO}}(x, y_i) = \frac{1}{G-1}\sum_{j \neq i} R(x, y_j). \tag{4}$$

The *raw* advantage for sample $y_i$ is then $A(x, y_i) = R(x, y_i) - b_{\text{GRPO}}(x)$, then it is normalized with the standard deviation $\sigma_G$. While simple to implement, this approach suffers from two key limitations. First, it is sample-inefficient, requiring $G > 1$ generations per prompt for each gradient step. Second, the baseline is estimated from a very small group ($G$), making it a high-variance estimate of the true value function, which in turn leads to noisy advantage estimates.

## 4 METHOD

We introduce Single-stream Policy Optimization (SPO), a method designed for policy optimization in settings with verifiable feedback (RLVR) (Lambert et al., 2024). We assume the feedback is binary[1], i.e., $+1$ for success and $0$ for failure. SPO addresses the challenge of estimating a non-stationary success probability for a policy that evolves over training iterations. It integrates a Bayesian value tracker with an adaptive memory mechanism into a policy gradient framework. The core components are: (1) a KL-adaptive tracker that provides a low-variance, single-sample estimate of the success probability; (2) a global advantage normalization scheme that ensures high sample efficiency and stable learning dynamics; and (3) prioritized sampling across training prompts to focus on prompts with high learning potential. The following subsections detail each component.

---

[1]Generalizing to non-binary rewards is straightforward, as discussed at the end of Section 4.1.

## 4.1 A KL-ADAPTIVE VALUE TRACKER

The definition of a value function is the *expected* reward of the prompt $x$ under policy $\pi$, i.e., $V_\pi(x) = \mathbb{E}_{y \sim \pi(\cdot|x)}[R(x,y)]$. We use $\hat{v}(x)$ to denote the tracker's running estimate of $V_\pi(x)$; that is, $\hat{v}(x) \approx V_\pi(x)$. To estimate the non-stationary success probability of a prompt $x$, we use a Bayesian *tabular* tracker instead of a separate value network[2]. For the binary success/failure rewards common in RLVR, this is elegantly modeled using a Beta distribution, which is the conjugate prior for the Bernoulli process governing the outcomes. We therefore model the success probability $\hat{v}(x)$ using a Beta distribution: $\hat{v}(x) \sim \text{Beta}(\alpha(x), \beta(x))$, where the value estimate is the posterior mean $\hat{v}(x) = \alpha(x)/(\alpha(x) + \beta(x))$.

The tracker adapts to policy changes by dynamically adjusting its memory of past rewards. When the policy changes significantly, older observations become less relevant and should be downweighted. After each new observation $r(x,y) \in \{0,1\}$, we discount the prior Beta parameters $(\alpha_{-1}, \beta_{-1})$ by a factor $\rho(x)$ before incorporating the new evidence $r(x,y)$:

$$\alpha(x) = \rho(x)\alpha_{-1}(x) + r(x,y), \quad \beta(x) = \rho(x)\beta_{-1}(x) + (1 - r(x,y)), \quad \hat{v}(x) = \frac{\alpha(x)}{\alpha(x) + \beta(x)}. \quad (5)$$

The discount factor $\rho(x) = 2^{-D(x)/D_{\text{half}}}$ is determined by the KL divergence $D(x)$ between the current policy and the last policy that *acted* on prompt $x$, causing the tracker to forget faster as the policy changes more significantly. The hyperparameter $D_{\text{half}}$ controls this forgetting rate $\rho \in [\rho_{\min}, \rho_{\max}]$.

**Initialization.** To initialize, we collect $n_0$ samples to compute an initial value estimate $\hat{v}_0(x)$. To avoid transient instability, we set the initial effective sample size to its expected equilibrium, $N_0 = 1/(1 - \rho_{\min})$, where $\rho_{\min}$ is the minimum allowed forgetting factor. The initial parameters are then:

$$\alpha_0(x) = N_0 \cdot \hat{v}_0(x), \qquad \beta_0(x) = N_0 \cdot (1 - \hat{v}_0(x)). \quad (6)$$

This Bayesian update is equivalent to an adaptive Exponential Moving Average (EMA) on the value estimate:

$$\hat{v}(x) = \hat{v}_{-1}(x) + \eta(x)(r(x,y) - \hat{v}_{-1}(x)), \quad (7)$$

where the learning rate $\eta(x) = (\rho(x)N_{\text{eff},-1}(x) + 1)^{-1}$ naturally adapts to both policy shifts (via $\rho(x)$) and statistical confidence (via $N_{\text{eff}} = \alpha(x) + \beta(x)$). This formulation highlights how our tracker balances new evidence against accumulated knowledge. For *general rewards* beyond binary ones, we can just use the same EMA formulation to directly track $\hat{v}$, rather than relying on $\alpha$ and $\beta$ in the binary cases.

## 4.2 ADVANTAGE ESTIMATION AND POLICY OPTIMIZATION

SPO uses the tracker's estimate $\hat{v}$ as a baseline for advantage calculation in a policy gradient algorithm. At iteration $i$, for a single reward $r(x,y)$ obtained with policy $\pi_{\theta_i}$, the advantage is computed using the *pre-update* baseline (denoted with subscript $_{-1}$):

$$A(x,y) = r(x,y) - \hat{v}_{-1}(x). \quad (8)$$

Using the baseline from the previous step ensures that it is independent of the action taken at step $i$, preserving the unbiasedness of the policy gradient estimate. While the reward $r(x,y)$ is typically a direct outcome signal, SPO's framework is also compatible with more sophisticated reward functions. For instance, recent work like InfAlign (Balashankar et al., 2024) demonstrates how to calibrate and transform the reward signal to be "inference-aware," directly optimizing for procedures like Best-of-$N$ sampling. Such transformed rewards can be seamlessly integrated into SPO by replacing the standard $r(x,y)$ in the advantage calculation. Since $v_{-1}(x)$ is independent of $y \sim \pi_{\theta_i}(\cdot|x)$, $\mathbb{E}[(r - v_{i-1}(x))\nabla_\theta \log \pi] = \nabla J(\theta)$ (Williams, 1992). Instead of normalizing advantages on a per-prompt basis in a group (Shao et al., 2024; DeepSeek, 2025), SPO normalizes them across an entire batch of prompts $\mathcal{B}$ (Hu et al., 2025; Schulman et al., 2017; Andrychowicz et al., 2020; Liu et al., 2025a). The normalized advantage $\tilde{A}(x,y)$ is computed as:

$$\tilde{A}(x,y) = \frac{A(x,y) - \mu_\mathcal{B}}{\sigma_\mathcal{B}}, \quad (9)$$

---

[2]The development of core RL algorithms was on tabular representation (Sutton & Barto, 2018).

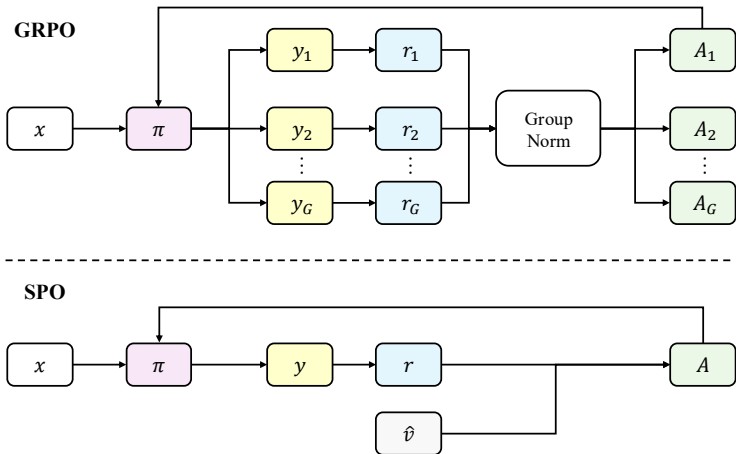

Figure 1: Illustrations of GRPO and SPO.

where $\mu_\mathcal{B}$ and $\sigma_\mathcal{B}$ are the mean and standard deviation of advantages in the batch $\{A(x,y)\}_{x \in \mathcal{B}}$. We then apply the advantage $\tilde{A}(x,y)$ to each *token* in the response sequence $y$ and update the policy parameters using a standard PPO-Clip policy loss (Schulman et al., 2017)[3]:

$$L^{\text{CLIP}}(\theta) = \mathbb{E}_{s,t}\left[\min\left(\frac{\pi_\theta(a_t \mid s_t)}{\pi_{\theta_{\text{old}}}(a_t \mid s_t)}\,\tilde{A},\; \text{clip}\left(\frac{\pi_\theta(a_t \mid s_t)}{\pi_{\theta_{\text{old}}}(a_t \mid s_t)},\, 1-\varepsilon,\, 1+\varepsilon\right)\tilde{A}\right)\right]. \quad (10)$$

Methods like Clip-Higher (Yu et al., 2025), Clip-Cov (Cui et al., 2025) and KL-Cov (Cui et al., 2025) to retain policy entropy are applicable here. Other policy optimization algorithms like CISPO (Mini-Max, 2025) (similar to `vtrace` (Espeholt et al., 2018; Wu et al., 2025)) and GSPO (Zheng et al., 2025a) (use sequence-level likelihood instead of token-level) are compatible with our advantage estimator. Advanced methods to control policy behaviors like ASPO (Lin & Xu, 2025) can be utilized to modulate the advantage values. We note that if we use "no baseline" (i.e., $\hat{v} = 0$), it is an extremely simple and valid algorithm but may suffer from high policy gradient variance.

### 4.3 PRIORITIZED PROMPT SAMPLING

---

**Algorithm 1** Single-stream Policy Optimization

---

1: **for** iteration $i = 1, 2, \ldots, T$ **do**
2:      For each $x \in \mathcal{X}$, compute sampling weight $w_i(x)$ according to Eqn. (11).
3:      Sample a batch of $B$ prompts $\mathcal{B}_i \subset \mathcal{X}$ according to weights $\{w_i(x)\}$.
4:      $\mathcal{D} \leftarrow \emptyset$
5:      **for** each prompt $x \in \mathcal{B}_i$ **do**
6:          Sample action $y \sim \pi_{\theta_{i-1}}(\cdot \mid x)$ and observe reward $r(x,y) \in \{0,1\}$.
7:          Compute raw advantage $A(x,y) \leftarrow r(x,y) - \hat{v}_{-1}(x)$.
8:          Store $(x, y, A(x,y))$ in $\mathcal{D}$.
9:          Update tracker $\hat{v}(x)$.
10:      Normalize advantages: $\tilde{A}(x,y) \leftarrow \big(A(x,y) - \mu_{\mathcal{B}_i}\big)/\sigma_{\mathcal{B}_i}$.
11:      Update $\theta_{i-1}$ to $\theta_i$ using mini-batches with a policy gradient algorithm (e.g., PPO-Clip).

---

To further enhance data efficiency, SPO employs a curriculum learning strategy by prioritizing prompts with the highest learning potential (Schaul et al., 2015; Sutton & Barto, 2018). At each iteration, we

---

[3]The term "PPO" is frequently used with ambiguity. It may denote the entire algorithm suite (e.g., clipped policy and value losses), refer narrowly to just the clipped policy objective, or describe the broader training framework, including mechanisms like mini-batch updates.

sample a batch of prompts based on a score that emphasizes prompts with high uncertainty, while ensuring a minimum level of exploration. The sampling weight $w_i(x)$ for prompt $x$ is defined as:

$$w_i(x) \propto \frac{\sqrt{\hat{v}_{-1}(x)\big(1 - \hat{v}_{-1}(x)\big)}}{N_{\text{eff},-1}^{\gamma}(x)} + \epsilon. \tag{11}$$

The first term corresponds to the estimated standard deviation of a Bernoulli outcome, which naturally allocates more weight to prompts that are neither almost always solved ($\hat{v} \approx 1$) nor almost always failed ($\hat{v} \approx 0$). The exploration bonus $\epsilon$, set to $0.05$ by default, prevents curriculum collapse by ensuring that every prompt retains a non-zero probability of being sampled, thereby maintaining broad coverage of the data distribution.

Here, ($N_{\text{eff}} = \alpha + \beta$) denotes the effective sample size of the Beta tracker and ($\gamma \in [0,1]$) controls how strongly we down-weight prompts with well-estimated success probabilities, allowing the sampler to balance value-uncertainty, driven prioritization against broader exploration.

The complete SPO training procedure is outlined in Algorithm 1.

## 4.4 ADVANTAGES OVER GRPO

**Group-Free for Scalable Infrastructure.** SPO's design is inherently "group-free", a significant advantage in distributed training frameworks for LLMs. Each sample, consisting of a single stream of (prompt, response) pair, is a self-contained data point for the policy update. GRPO, however, requires the generation and evaluation of an entire group of $G$ samples for a single prompt before any training signal can be computed. We provide our illustrations in Figure 1. In a distributed setting, this introduces a synchronization barrier: the processing of a given prompt is not complete until all $G$ responses have been generated. This is particularly problematic in the presence of long-tail generation times, where a single slow response generation can stall the processing for its *entire group*. For constructing a training batch, SPO only needs to collect $B$ independent (prompt, response) pairs, which is far more flexible and efficient than waiting for $B$ entire groups to complete. This makes SPO's architecture significantly more infrastructure-friendly and scalable. The advantage is amplified in agentic training, especially in settings that require multi-turn interactions with tools (Gao et al., 2025; Chen et al., 2025b) or long-horizon agent rollouts (Zhipu, 2025; Xu et al., 2025). The scale of these interactions can be substantial: state-of-the-art open-source models (`gpt-oss-120b`) may average 20 search turns per task (Chen et al., 2025b), with other agentic sessions reaching over 40 tool calls and generating up to 150,000 tokens of context (Gao et al., 2025).

**Adaptive Curriculum.** To further enhance training efficiency, SPO integrates a prioritized sampling scheme. This mechanism naturally creates an adaptive curriculum by focusing computational resources on prompts with the highest learning potential. This ensures that the model's training is concentrated on the most informative examples at any given point in time. GRPO, in its standard formulation, typically relies on uniform sampling of prompts. This may waste computation on prompts that are already mastered or are currently too difficult to yield useful learning signals. While dynamic sampling (Yu et al., 2025) and repeat strategies (An et al., 2025) have been proposed to mitigate this issue, they often discard samples *after* generation, wasting computation. SPO's prioritized sampling addresses the scheduling problem *before* response generation, leading to a more natural and efficient training process.

More discussions on the *inefficiency* of dynamic sampling and the *variance reduction* of policy gradient are outlined in Appendix D, where we provide detailed analysis.

## 5 EXPERIMENTS

### 5.1 EXPERIMENTAL SETUP

The SPO algorithm is broadly applicable in LLM reasoning tasks (DeepSeek, 2025) and Agentic training. We evaluate Tool-Integrated Reasoning (TIR) (Feng et al., 2025; Lin & Xu, 2025) scenarios, where the LLMs can utilize external Python interpreter to help solve hard problems. We conduct experiments using a moderately sized LLM, Qwen3-8B (Qwen, 2025). For training data, we use

Table 1: Comparison of GRPO and SPO on five benchmarks using maj@32 and avg@32. Averages are shown in the last column. Bold indicates the better-performing method for each metric.

| Method | AIME 24 | | AIME 25 | | BeyondAIME | | BRUMO 25 | | HMMT 25 | | Average | |
|---|---|---|---|---|---|---|---|---|---|---|---|---|
| | maj@32 | avg@32 | maj@32 | avg@32 | maj@32 | avg@32 | maj@32 | avg@32 | maj@32 | avg@32 | maj@32 | avg@32 |
| Qwen3-8B | 77.8 | 64.4 | 70.5 | 58.4 | 45.2 | 38.0 | 55.1 | 49.4 | 36.8 | 30.3 | 57.1 | 48.1 |
| GRPO | 83.3 | **77.6** | 72.1 | 64.2 | 45.6 | 39.0 | 56.7 | 56.9 | 44.2 | **40.9** | 60.4 | 55.7 |
| SPO | **84.0** | 74.9 | **76.5** | **65.0** | **46.9** | **40.3** | **64.0** | **59.0** | **47.5** | 40.6 | **63.8** | **56.0** |

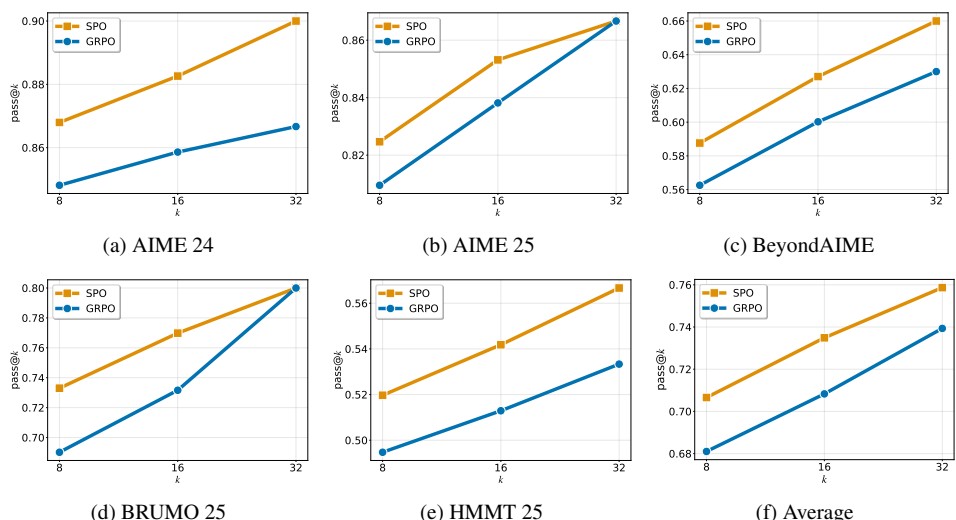

(a) AIME 24     (b) AIME 25     (c) BeyondAIME

(d) BRUMO 25     (e) HMMT 25     (f) Average

Figure 2: Pass@$k$ plots comparing GRPO and SPO across five math competition benchmarks.

the English subset from the DAPO dataset (Yu et al., 2025). Only outcome reward is applied for RLVR, without the format rewards. We evaluate performance on the challenging math competition benchmarks, i.e., AIME 24, AIME 25, BeyondAIME (Seed, 2025), BRUMO 25 (Balunović et al., 2025), and HMMT 25 (Balunović et al., 2025). See Appendix E for training and evaluation details.

We distinguish our goal from that of "hill-climbing" on benchmark leaderboards. The latter often necessitates resource-intensive and highly specialized techniques, including SFT from frontier models (Liu et al., 2025b), mid-training (Wang et al., 2025), multi-stage RL pipelines (Luo et al., 2025; He et al., 2025; Chen et al., 2025a), curated hard datasets with intricate processing (An et al., 2025; Shang et al., 2025), test-time scaling techniques (Fu et al., 2025) and extremely large generation group sizes (Zhipu, 2025). Our work, instead, concentrates on the fundamental efficiency and scalability of the RL algorithm itself.

## 5.2 EMPIRICAL COMPARISON WITH GRPO

Our experiments demonstrate that SPO outperforms the GRPO baseline on aggregate metrics when training the Qwen-8B model. As shown in Table 1, SPO achieves superior weighted average scores on both primary metrics. It obtains a maj@32 of 63.8 compared to GRPO's 60.4, a significant improvement of $+3.4$ percentage points (pp). This aggregate strength is driven by remarkable consistency, as SPO outperforms GRPO on the maj@32 metric across all five benchmarks. The performance gap is most pronounced on **BRUMO 25**, where SPO achieves a substantial $+7.3$ pp (64.0 vs. 56.7). Further significant gains are seen on **AIME 25** ($+4.4$ pp) and **HMMT 25** ($+3.3$ pp points), underscoring the robustness of SPO's improvements. Notably, these benchmarks have minimal data contamination (Balunović et al., 2025), allowing them to serve as a true test of *generalization*. This demonstrates that our SPO method improves the model's ability to generalize rather than simply overfit to the training data, a risk exemplified by the DAPO dataset's strong correlation with AIME 24. While GRPO remains competitive on the avg@32 metric in some cases, SPO's consistent and significant advantage in maj@32 suggests it learns more robust and repeatable solutions, a key goal for reliable reasoning models.

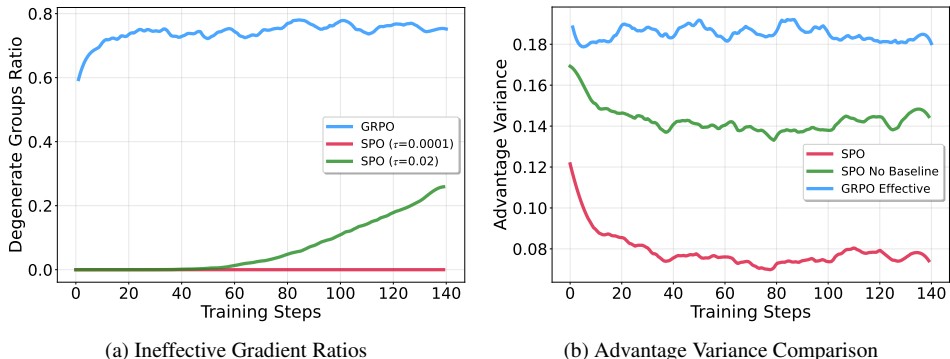

(a) Ineffective Gradient Ratios

(b) Advantage Variance Comparison

Figure 3: Signal Efficiency and Stability Analysis of SPO *vs.* GRPO. (a) GRPO suffers from a high ratio of degenerate groups (blue), which yield no learning signal. In contrast, SPO's rate of near-zero advantages (red/green) increases as the model learns, reflecting prediction accuracy rather than wasted computation. (b) SPO's baseline (red) provides a stable, low-variance signal, significantly reducing the raw reward variance (green). GRPO's effective advantage (blue), calculated only on non-degenerate samples, is highly volatile and unstable.

These findings are mirrored in the pass@$k$ performance shown in Figure 2. The weighted average curve (Figure 2f) shows a clear and consistent advantage for SPO across all values of $k$, translating to an average improvement of approximately $2.4$ pp. While the performance on avg@32 is more competitive on a per-benchmark basis, SPO's strong overall performance underscores the stability and effectiveness of its learning signal. We provide additional ablation studies on $A^*$-PO, SPO with no baseline, and SPO with no offline initialization in Appendix F.

## 5.3 ANALYSIS OF SIGNAL EFFICIENCY AND STABILITY

To empirically assess the architectural advantages of SPO, we conduct a two-part analysis of the unnormalized advantage signals produced by SPO and GRPO (Figure 3). First, we quantify complete signal loss arising from degenerate groups. Second, we measure the variance of the remaining learning signals. Together, these metrics characterize each method's efficiency and stability.

**Signal Efficiency and Information Loss.** Figure 3a reports the fraction of ineffective samples. For GRPO (blue), the share of samples in degenerate groups rises from roughly 60% to over 80%, yielding zero advantage and no gradient. For SPO, we instead track the proportion of near-zero advantages under two diagnostic tolerances, $|A| \leq \tau$, with values of $\tau = 10^{-4}$ (red) and $\tau = 0.02$ (green). Advantages under the tight tolerance $\tau = 10^{-4}$ remain rare throughout training (red line), while the $|A| \leq 0.02$ share (green) gradually increases as the value tracker $\hat{v}$ becomes more accurate and residuals shrink on mastered prompts. This trend is expected and desirable: it reflects accurate prediction rather than signal loss. Unlike GRPO's degenerate groups, these SPO samples are not discarded, they still produce well-defined gradients and contribute to learning. Notably, even under the stricter $\tau = 0.02$ tolerance, SPO's near-zero ratio remains far below GRPO's degenerate rate, underscoring SPO's efficient use of compute.

**Signal Stability and Advantage Variance.** Figure 3b compares advantage variance across methods. As a reference, the green line ("SPO No Baseline") corresponds to raw rewards, i.e., the high-variance signal faced by vanilla policy gradient. SPO's history-informed baseline (red) delivers a substantial, stable variance reduction of nearly 50%. For GRPO, computing variance only over non-degenerate samples ("GRPO Effective", blue) reveals a highly volatile signal with the largest variance among all conditions, exceeding even "SPO No Baseline". We conclude that SPO's baseline is effective, yielding stable, low-variance gradients, whereas GRPO's on-the-fly baseline is noisy and destabilizing when it produces a signal. The apparent stability of GRPO's overall variance is driven by the prevalence of zero-variance degenerate samples and thus reflects inefficiency rather than robustness.

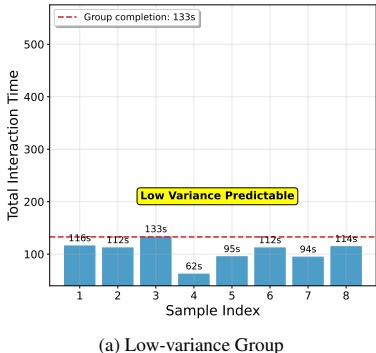 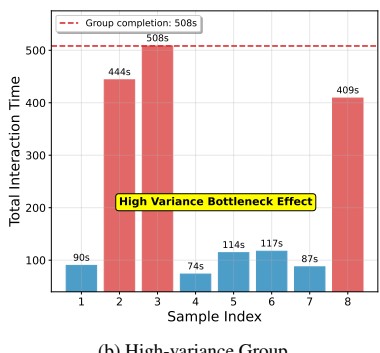

(a) Low-variance Group  (b) High-variance Group

Figure 4: The Bottleneck Effect in Group-Based Sampling. (a) In a low-variance environment, sample completion times are predictable, and the group synchronization cost is minimal. (b) In a realistic high-variance agentic environment, three slow trajectories ($444s$, $508s$, and $409s$) create a severe bottleneck, forcing the entire group to wait and wasting the compute used for the six faster samples.

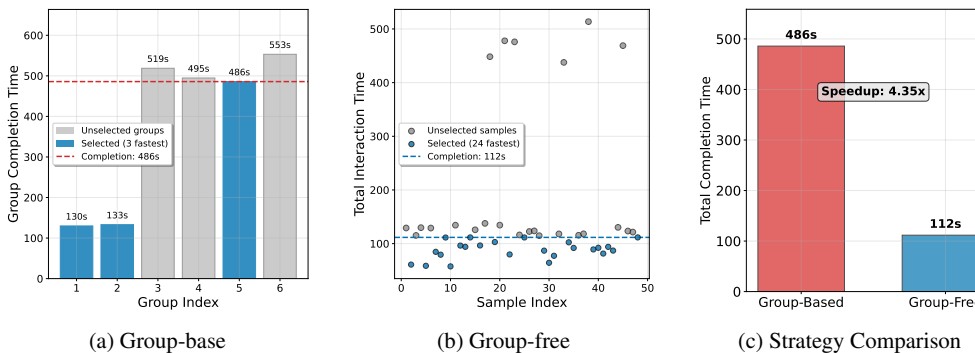

(a) Group-base  (b) Group-free  (c) Strategy Comparison

Figure 5: Throughput Comparison: Group-Based vs. Group-Free. (a) A group-based strategy, even when parallelized, is bottlenecked by its slowest group, taking $486s$ to collect a batch of 3 groups (24 samples). (b) A group-free strategy collects the 24 fastest samples from a larger pool of 48, completing the batch in just $112s$ by avoiding stragglers. (c) The group-free approach achieves a $4.35\times$ speedup, demonstrating its superior efficiency for agentic training.

## 5.4 AGENTIC TRAINING DEMONSTRATIONS

We perform simulations to demonstrate the practical implications of SPO's group-free design in agentic training scenarios, where interaction times can be highly variable. Group-based methods like GRPO suffer from a critical scalability bottleneck due to their inherent synchronization barrier, a problem that is particularly acute in agentic tasks involving multi-turn tool use or long-horizon reasoning.

Figure 4 illustrates this fundamental issue. In an idealized low-variance setting (Figure 4a), where all agentic trajectories complete in similar times, the group-based approach is efficient. However, in a more realistic high-variance setting (Figure 4b) characterized by long-tail latencies, a single slow-running trajectory (a "straggler") can stall the entire group. In our simulation, while most samples finish in under $133$ seconds, the group must wait $508$ seconds for its slowest member. This bottleneck effect forces faster samples to remain idle, severely hindering training throughput and wasting computational resources.

SPO's group-free architecture directly resolves this inefficiency. Figure 5 compares the time required to assemble a training batch of 24 samples using both strategies. The group-based approach (left), even when optimized by running 6 groups in parallel and selecting the 3 fastest, is still constrained by the slowest trajectory within those selected groups, taking $486s$ to complete. In contrast, the group-free approach (middle) leverages asynchrony by starting 48 independent samples and simply collecting the first 24 to finish. In our simulated scenario, this process takes only $112s$, as it naturally

filters out the slow outliers. As shown on the right, this architectural difference results in a significant $4.35\times$ speedup in this realistic agentic simulation. Simulations show that SPO's architecture can lead to significant throughput gains, making it a more scalable and robust foundation for training on complex, long-horizon agentic tasks.

## 6 CONCLUSIONS

We identified critical inefficiencies in group-based policy optimization methods for LLMs, namely computational waste from degenerate groups and scalability bottlenecks from synchronization. To address these, we proposed Single-stream Policy Optimization (SPO), a principled return to the classic single-stream paradigm. SPO replaces the noisy, per-group baseline with a persistent KL-adaptive value tracker and global advantage normalization, creating a more stable and efficient learning signal.

Our empirical results demonstrate that SPO's design is not merely simpler, but superior. It consistently outperformed GRPO on complex reasoning tasks while eliminating the systemic flaws of its group-based counterpart. By demonstrating that a well-designed single-stream approach can surpass more complex methods, our work challenges the prevailing trend of adding incidental complexity to RL algorithms for LLMs. SPO provides a robust, scalable, and efficient foundation for future research in agentic and reasoning model training, highlighting the enduring power of foundational reinforcement learning principles. Future work can focus on refining the best practices for applying SPO and exploring its limits, pushing its effectiveness to power the next generation of reasoning and agentic LLMs.

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

## A    LLMs Usage

LLMs were employed solely to assist with minor editorial improvements to the text. Their use was limited to polishing language, and they did not contribute to the conceptual or scientific content of the paper.

## B    SPO Initialization

We show the SPO initialization procedure in Algorithm 2. In the experiments, we use $n_0 = 8$ to have a good estimation of initial baseline tracker. We ablate the setting where we use *no offline estimation* and rely on the online moving estimator in Section F.

---
**Algorithm 2** SPO Initialization

---
 1: Set initial effective sample size $N_0 = 1/(1 - \rho_{\min})$.
 2: **for** each prompt $x \in \mathcal{X}$ **do**
 3:     Collect $n_0$ outcomes $\{r^{(k)}\}_{k=1}^{n_0}$ with an initial policy $\pi_0$.
 4:     Compute initial value estimate $\hat{v}_0(x) = \frac{1}{n_0} \sum_{k=1}^{n_0} r^{(k)}$.
 5:     Set $\alpha_0(x) = N_0 \cdot \hat{v}_0(x)$ and $\beta_0(x) = N_0 \cdot (1 - \hat{v}_0(x))$.

---

Practically, one may concern about the extra cost during the offline estimation of $\hat{v}_0$. We note that we share the offline estimation for our experiments so that people could skip this process and directly load our datasets, and there are datasets like Polaris (An et al., 2025) that pre-compute accuracy for Deepseek-R1-Distill-Qwen-7B (DeepSeek, 2025). The cost can be *amortized* across the experiments people run themselves, and we will share more `(dataset, base_model)` combinations to facilitate experiment efficiency.

## C    Batch Extensions

We could adapt Single-stream Policy Optimization (SPO) into a prompt-repetition scheme[4], processing each prompt $G$ times per batch with a shared baseline estimator $\hat{v}$ to better handle sparse rewards. Our method's primary advantage over GRPO lies in its asynchronous nature, achieved by removing the group synchronization barrier. Treating repeated prompts as independent trajectories unlocks two key efficiency improvements. First, it enables robust handling of long-tail generation issues, as slow or problematic trajectories can be terminated early, discarded, or managed via partial rollouts (Kimi, 2025) without delaying the entire batch. Second, it facilitates a more flexible batching strategy. By over-sampling the number of initial prompts (e.g., by 50%), a full training batch can be assembled from the first-finishing trajectories, allowing the optimization step to proceed immediately without waiting for stragglers. This design significantly reduces training latency compared to the rigid group synchronization required by GRPO. When tackling hard prompts, the batch extensions may help obtain learning signals more quickly.

## D    Comparisons against GRPO

### D.1    Inefficiency of Dynamic Sampling

To address the information loss from degenerate sample groups (where all rewards are identical), methods like DAPO (Yu et al., 2025) employ *dynamic sampling*. This strategy continues generating responses for a prompt until the collected set contains at least one success and one failure, guaranteeing a non-zero advantage. While effective at ensuring a learning signal, this approach can be extremely data- and time-inefficient. Note that when people report performance with dynamic sampling, the "steps" indicate the *learning* steps rather than the *sampling* steps, where the latter is normally a multiple of the former (e.g., $5\times$).

---
[4]Batch SPO or BSPO

We can formalize the expected computational cost. For a prompt $x$ with true success probability $p = V_\pi(x)$, let $N$ be the number of samples required to obtain a non-degenerate set. We have:

$$\mathbb{E}[N \mid p] = p\left(1 + \tfrac{1}{1-p}\right) + (1 - p)\left(1 + \tfrac{1}{p}\right) = \frac{1}{p(1 - p)} - 1.$$

This cost grows hyperbolically as the policy becomes either proficient ($p \to 1$) or incompetent ($p \to 0$). For example, if a policy has a 10% success rate ($p = 0.1$), the expected number of generations needed to collect both a success and a failure is $\mathbb{E}[N] \approx 10.11$. In contrast, SPO requires exactly one sample per prompt and uses its adaptive curriculum to actively *de-prioritize* these inefficient prompts, allocating resources to where learning is most effective. This makes SPO fundamentally more scalable and computationally efficient.

## D.2 Variance Reduction for Policy Gradient

The per-sample policy gradient is $g = A(x, y)\nabla_\theta \log \pi_\theta(y|x)$, where the advantage $A$ is an estimate of the expected return over a baseline. The variance of this gradient, $\text{Var}[g]$, is a key driver of training efficiency. We analyze how the construction of the advantage $A$ leads to significant variance differences between GRPO and SPO.

**GRPO's High-Variance Group-Based Advantage**: GRPO computes advantages by comparing outcomes within a small group of $G$ ($G = 8, 16, ...$) samples generated for the same prompt. The normalized advantage for sample $x$ with binary reward $r \in \{0, 1\}$ is $\tilde{A}_{\text{GRPO}} = \frac{r - \mu_{\mathcal{G}}}{\sigma_{\mathcal{G}} + \epsilon}$, where both the baseline $\mu_{\mathcal{G}}$ (e.g., the group mean $\frac{1}{G}\sum_j r_j$) and the standard deviation $\sigma_{\mathcal{G}}$ are estimated from the same small group of $G$ samples. This coupled, small-sample estimation introduces three fundamental sources of variance:

- **Noisy Baseline (Numerator):** The baseline $\mu_{\mathcal{G}}$, estimated from only $G$ samples, where $G$ is small, is a high-variance quantity. This inflates the variance of the unnormalized advantage $(r - \mu_{\mathcal{G}})$ by a factor of $(1 + \frac{1}{G})$ compared to using an optimal baseline.

- **Noisy Scaling (Denominator):** The standard deviation $\sigma_{\mathcal{G}}$, estimated from only $G$ samples, is also highly variable. Scaling the gradient by this noisy random variable further increases total variance.

- **Information Loss (Degeneracy):** When all rewards in the group are identical (e.g., all 0s or all 1s), the advantage for every sample becomes zero, providing no gradient signal. This event, which occurs with probability $Z_G(p) = p^G + (1-p)^G$ where $p = V^\pi(x)$, effectively reduces the batch size and inflates variance by a factor of $1/(1 - Z_G(p))$, an issue that is especially severe for easy ($p \approx 1$) or hard ($p \approx 0$) prompts.

**SPO's Low-Variance Decoupled Advantage**: In contrast, SPO is designed to minimize these variance sources by decoupling the advantage calculation from the current group of samples. It uses an action-independent baseline $b = \hat{v}(x)$ from a historical tracker, which provides a stable, low-variance estimate of the true success probability $p$. The advantage is simply $A_{\text{SPO}} = \texttt{batch\_norm}(r(x, y) - \hat{v}(x))$. Crucially, SPO then applies *global* normalization (Schulman et al., 2017; Andrychowicz et al., 2020; Liu et al., 2025c), scaling all advantages in a large batch of size $B \gg G$ by a single, stable standard deviation $\sigma_{\text{batch}}$. This design avoids GRPO's pitfalls: the baseline $b$ is near-optimal, the normalization scaler $\sigma$ is stable, and there is no systematic information loss from group-outcome degeneracy.

**Quantitative Comparison**: A simplified ratio of the reward-term variance quantifies the difference:

$$\frac{\text{Var}[g]_{\text{GRPO}}}{\text{Var}[g]_{\text{SPO}}} \approx \underbrace{\frac{1 + \frac{1}{G}}{1 + \frac{1}{N_{\text{eff}}+1}}}_{\text{Baseline Noise}} \times \underbrace{\frac{1}{1 - Z_G(p)}}_{\text{Information Loss}} \times \underbrace{\frac{1 + \psi_G}{1 + \psi_{\mathcal{B}}}}_{\text{Normalization Noise}} . \tag{12}$$

Here, $N_{\text{eff}}$ is the effective sample count for SPO's tracker, and $\psi_G > 0$ captures the excess variance from per-group, $\psi_{\mathcal{B}}$ represents the excess variance introduced by estimating the normalization statistics (mean and standard deviation) from a large global batch of size $N_{\mathcal{B}}$ ($\psi_{\mathcal{B}} \approx 0$). For a moderately difficult prompt ($p = 0.5$) with $G = 8$, the normalization noise dominates. However, for an easy/hard prompt ($p = 0.9/p = 0.1$), the information loss term dominates, and the ratio swells to

$\approx 1.97$. While increasing $G$ in GRPO mitigates information loss, it does so at a multiple generation cost and cannot fix the inherent noise from its small-sample baseline and scaling. SPO achieves lower variance more efficiently by design.

## E TRAINING AND EVALUATION DETAILS

All experiments in this paper are implemented on top of verl (Sheng et al., 2024) and ReTool (Feng et al., 2025) for the tool-integrated reasoning setup. During training, we set the maximum response length to 16,384 tokens. The policy learning rate is fixed at $1 \times 10^{-6}$. Following DAPO (Yu et al., 2025), we adopt the Clip-Higher mechanism, with clipping parameters $\varepsilon_{\text{low}} = 0.2$ and $\varepsilon_{\text{high}} = 0.28$, to balance exploration and exploitation. The sampling parameters are set to temperature 1.0, top-$p = 1.0$, and top-$k = -1$. The forgetting rate thresholds are chosen as $\rho_{\min} = 0.875$ and $\rho_{\max} = 0.96$, yielding window sizes $W_{\min} = 1 - \frac{1}{\rho_{\min}} = 8$ and $W_{\max} = 25$.

GRPO rollouts are collected with multiple responses per prompt, and training mini-batch sizes are chosen such that 8 gradient updates are performed per rollout step. For a fair comparison, the prompt batch size in SPO is set equal to the total number of responses in GRPO, as SPO generates only a single response for each prompt. Specifically, GRPO uses a prompt batch size of 256 with 8 responses per prompt and a training mini-batch size of 256, while SPO operates on $2,048 = 256 \times 8$ prompts. Both algorithms are set with maximum of 8 Python interpreter interaction turns.

For evaluation on hard math competition benchmarks, i.e., AIME 24, AIME 25, BeyondAIME (Seed, 2025), BRUMO 25 (Balunović et al., 2025) and HMMT 25 (Balunović et al., 2025), we set sampling parameters to temperature 0.6, top-$p$ 0.95, and top-$k$ 20, as officially recommended. We define a binary reward function $r_{i,j}$ such that a response receives $r_{i,j} = 1$ if the final answer is correct, and $r_{i,j} = 0$ otherwise. The same reward function is consistently used during training for policy optimization and during evaluation. We set the maximum response token to 32,768.

Given a test set with $M$ problems, and for each problem $i$ we independently sample $k$ responses with rewards $\{r_{i,1}, r_{i,2}, \ldots, r_{i,k}\}$, we define:

- **avg@$k$**: the expected correctness of an individual response:

$$\text{avg@}k = \frac{1}{M} \sum_{i=1}^{M} \frac{1}{k} \sum_{j=1}^{k} r_{i,j}.$$

- **pass@$k$**: the probability of solving a problem within $k$ attempts. Directly computing $\mathbf{1}\left(\max_{1 \leq j \leq k} r_{i,j} = 1\right)$ can lead to high variance. Following (Chen et al., 2021), we instead generate $n \geq k$ responses per problem, count the number of correct ones $c \leq n$, and use the unbiased estimator:

$$\text{pass@}k = \frac{1}{M} \sum_{i=1}^{M} \left[ 1 - \frac{\binom{n-c_i}{k}}{\binom{n}{k}} \right],$$

where $c_i$ denotes the number of correct responses for problem $i$.

- **maj@$k$**: the correctness of the majority-voted answer (Wang et al., 2023). This metric first identifies the most frequent answer among $k$ responses for each problem. The score is 1 if that modal answer is correct, and 0 otherwise. Let $a_{i,j}$ be the final answer string for the $j$-th response to problem $i$, and let $r(\cdot)$ be the reward function for a given answer string. The metric is defined as:

$$\text{maj@}k = \frac{1}{M} \sum_{i=1}^{M} r\left(\text{mode}\{a_{i,j}\}_{j=1}^{k}\right).$$

## F ABLATION STUDIES

We conduct a series of ablation studies to dissect the core components of SPO and validate our design choices. To facilitate efficient experimentation, these studies are performed under a streamlined setting compared to our main experiments. Specifically, we utilize a batch size of 256 prompt-response pairs, and the model is updated with 4 gradient steps for each collected batch. All ablation results are

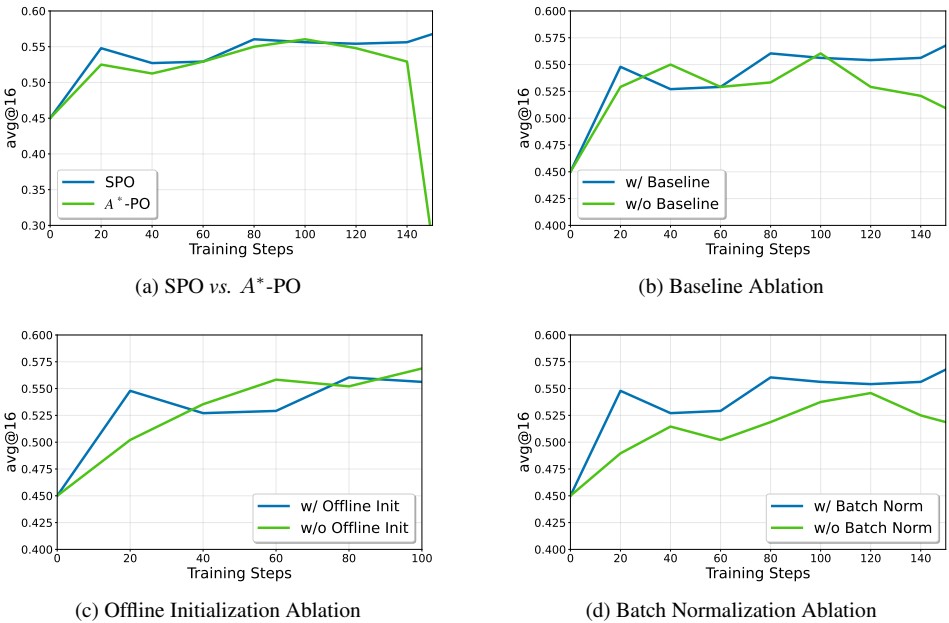

Figure 6: Ablation studies evaluating the core components of SPO. (a) SPO's adaptive baseline outperforms the static baseline of $A^*$-PO, demonstrating the benefit of a value function that evolves with the policy. (b) Removing the value tracker ("w/o Baseline") causes a severe performance drop, confirming its critical role in reducing gradient variance. (c) Eliminating the offline initialization step ("w/o Offline Init") leads to initial training instability and suboptimal convergence, highlighting the importance of a warm start for the value tracker. (d) Removing batch normalization ("w/o Batch Norm") results in slower convergence and reduced stability, showing the normalization's role in stabilizing updates.

reported on the AIME 25 benchmark, using the avg@16 metric with a maximum generation length of $16, 384$ tokens.

**SPO *vs*. $A^*$-PO**. This experiment, presented in Figure 6a, compares our proposed SPO with $A^*$-PO (Brantley et al., 2025). $A^*$-PO utilizes a static baseline derived from a pre-computed optimal value function, $V^*$, which is tied to the KL-regularized objective with respect to an initial reference policy, $\pi_{\text{ref}}$. While this approach is highly efficient, its central assumption may be challenged in tool-calling scenarios. In these tasks, learning involves acquiring new functional capabilities, leading to a significant policy drift where the learned policy, $\pi_t$, diverges substantially from $\pi_{\text{ref}}$. Consequently, the pre-computed $V^*$ may become a less representative baseline for the current policy's true value function, $V_{\pi_t}$, potentially affecting the accuracy of the advantage estimates. In contrast, SPO's baseline is adaptive, dynamically tracking an estimate of $V_{\pi_t}$ as the policy evolves. The empirical results, which show SPO's superior performance, suggest that this adaptability is crucial. By maintaining a baseline that remains relevant to the current policy, SPO provides a more stable and effective learning signal in environments that demand significant policy evolution. Finally, from a practical perspective, $\pi_{\text{ref}}$ computation during $A^*$-PO policy update occupies an extra trunk of GPU memory, making it less appealing than the proposed SPO algorithm.

**Baseline Ablation**. Figure 6b presents a crucial ablation that validates the fundamental principle of using a baseline for variance reduction. In this experiment, we remove the value tracker component $\hat{v}_{-1}(x)$ from the advantage calculation, causing the algorithm to rely solely on the globally batch-normalized raw reward $r(x, y)$ as its learning signal. However, the substantial performance degradation observed is a classic illustration of the remaining challenges. While global normalization effectively controls the overall scale of rewards, the raw reward signal is still noisy on a per-sample basis as it fails to account for prompt-specific difficulty. SPO's history-informed baseline is designed to subtract this expected difficulty, thereby effectively reducing variance and providing a much cleaner, more reliable gradient for learning. This experiment confirms that the adaptive value tracker

is the most critical component for SPO's success, directly addressing the core challenge of variance in single-stream policy optimization.

**Offline Initialization Ablation**. In Figure 6c, we analyze the impact of the value tracker's initialization phase. The standard SPO algorithm initializes the value tracker with estimates computed from a small set of $n_0$ offline samples, giving it a "warm start". The ablation removes this step, forcing the tracker to learn from scratch online. The results clearly demonstrate the benefit of the offline initialization. Without it, the tracker begins with a highly inaccurate baseline, leading to high-variance gradients and significant instability in the initial training phase, as evidenced by the performance dip. Although the model's performance is initially lower, it eventually recovers and catches up to the level of the properly initialized model. This initial performance gap nevertheless underscores the importance of a good initial value estimate for ensuring training stability and accelerating convergence.

**Batch Normalization Ablation**. Figure 6d isolates the contribution of SPO's global advantage normalization (Eqn. 9). In this variant, we compute policy updates using the unnormalized advantages $A(x, y) = r(x, y) - v_{-1}(x)$, omitting the batch centering and scaling. Compared to the full method, the training curve rises more slowly and exhibits larger oscillations, with a lower final avg@16. This behavior is consistent with our analysis: without standardizing by the batch statistics $(\mu_{\mathcal{B}}, \sigma_{\mathcal{B}})$, the effective step size becomes entangled with the transient reward scale and the prompt-mix in each batch, inflating gradient variance and worsening PPO clipping dynamics. In contrast, SPO's batch-wise normalization stabilizes the update scale across iterations, yielding faster and smoother convergence. Together with the baseline and offline-initialization ablations, these results underscore that SPO's low-variance learning signal derives from both a history-informed baseline and global, batch-level normalization, as predicted by the variance decomposition in Appendix D.2.

# G  ADDITIONAL EXPERIMENTS AND ANALYSES

This section reports additional experiments conducted after the initial submission. They probe the sensitivity of the KL-adaptive value tracker, the effect of the prompt curriculum, the robustness of the warm-start initialization, and the competitiveness of SPO against further baselines and model scales. Unless otherwise stated, all new studies are performed on Qwen3-8B with the Tool-Integrated Reasoning setup described in Appendix E, using the DAPO training corpus and the same reward and evaluation protocols as in the main text.

**Training setup.**    For the additional experiments we slightly reduce the per-step training budget due to time constraints. GRPO (with and without global normalization) and RLOO use a prompt batch size of 128, group size $G = 8$, and mini-batch size 16. SPO uses a single-stream batch of 1,024 prompts and a mini-batch size of 128, so that the total number of sampled trajectories per update matches the group-based methods (128 prompts × 8 samples). In all settings, the maximum response length is set to 8,192 tokens; other hyperparameters follow Appendix E.

## G.1  SENSITIVITY OF THE KL-ADAPTIVE VALUE TRACKER

**Incorporating effective sample size into the curriculum.**    The sampling weight in Eq. 11 prioritizes prompts according to the Bernoulli standard deviation $\sqrt{\hat{v}(1 - \hat{v})}$, which is maximized when $\hat{v} \approx 0.5$ and down-weights prompts that are almost always solved or failed. As pointed out in the reviews, this heuristic does not explicitly account for the estimation confidence encoded by the effective sample size $N_{\text{eff}} = \alpha + \beta$ of the Beta tracker.

To investigate this, we introduce an alternative sampling weight that down-weights prompts once their success probability is well-estimated:

$$w_i(x) \propto \frac{\sqrt{\hat{v}_{i-1}(x)\big(1 - \hat{v}_{i-1}(x)\big)}}{N_{\text{eff},i-1}(x)} + \epsilon, \tag{13}$$

where $N_{\text{eff},i-1}(x)$ is the effective sample size of the tracker for prompt $x$ before iteration $i$ and $\epsilon$ is the exploration bonus from Eq. 11. Table 2 compares the original and $N_{\text{eff}}$-aware curricula on AIME 24 and AIME 25 at $D_{\text{half}} = 0.08$.

Table 2: Effect of incorporating $N_{\text{eff}}$ into the sampling weight on AIME 24/25 maj@16 (Qwen3-8B, $D_{\text{half}} = 0.08$).

| Sampling weight | AIME 24 maj@16 | AIME 25 maj@16 |
|---|---|---|
| $\sqrt{\hat{v}(1-\hat{v})} + \epsilon$ | 55.95 | 52.75 |
| $\sqrt{\hat{v}(1-\hat{v})/N_{\text{eff}}} + \epsilon$ | 60.64 | 46.24 |

Table 3: Ablation on $D_{\text{half}}$ for Qwen3-8B with the $N_{\text{eff}}$-aware curriculum. Higher is better (maj@16).

| $D_{\text{half}}$ | AIME 24 maj@16 | AIME 25 maj@16 |
|---|---|---|
| 0.06 | 57.58 | 51.28 |
| 0.08 | 60.64 | 46.24 |
| 0.10 | 58.90 | 52.73 |
| 0.12 | **67.20** | **55.23** |

The mixed trend highlights a nuanced exploration–exploitation trade-off. Dividing by $N_{\text{eff}}$ encourages cycling through prompts more aggressively: once a prompt has been sampled many times, its weight decreases even when $\hat{v} \approx 0.5$. This broader coverage appears beneficial for AIME 24, but less so for AIME 25, where spending more steps on a smaller set of challenging prompts is advantageous. We therefore modify the sampling weight in the mainpaper to incorporate $N_{\text{eff}}$ in a manner that balances this need for exploration with the deep learning required for generalization.

**Sensitivity to $D_{\text{half}}$.** The hyperparameter $D_{\text{half}}$ controls how quickly the Beta tracker forgets past observations as the policy drifts: a larger $D_{\text{half}}$ yields a longer effective memory, reducing variance but increasing bias with respect to the current policy. We ablate $D_{\text{half}} \in \{0.06, 0.08, 0.10, 0.12\}$ using the $N_{\text{eff}}$-aware curriculum above. Results are summarized in Table 3.

Performance is relatively stable over a broad range, but $D_{\text{half}} = 0.12$ provides the best overall trade-off. In our setting (learning rate, batch size, and reward sparsity), this value implicitly defines a "half-life" that allows the tracker to accumulate enough history to reduce variance, while not retaining excessively stale information once the policy has moved far in KL.

### G.2 ANALYSIS OF THE PROMPT CURRICULUM

**Effect of prioritized sampling.** To isolate the contribution of the automatic curriculum, we compare the full SPO algorithm (with prioritized sampling from Eq. 11) against a variant without sampling mechanism, keeping all other components unchanged (including the value tracker and global normalization). Table 4 reports maj@16 on AIME 24 and AIME 25. Removing the curriculum leads to a substantial drop ($-10.8$ and $-9.3$ percentage points on AIME 24 and AIME 25 respectively), confirming that focusing computation on prompts with high estimated learning potential is critical for both data efficiency and final performance.

**Revisiting hard prompts.** A natural concern with the variance-based weight $\sqrt{\hat{v}(1-\hat{v})}$ is that prompts that are consistently failed ($\hat{v} \approx 0$) could effectively disappear from training. In Eq. (11) we explicitly prevent this collapse by adding an exploration bonus $\epsilon$, which ensures a non-zero lower bound on each prompt's sampling probability. As the model improves, such hard prompts are periodically revisited, allowing the tracker to update their value estimates if the policy's capabilities increase.

### G.3 WARM-START SAMPLE SIZE $n_0$

Appendix F already ablated the extreme case of removing the offline initialization entirely ($n_0 = 0$), which leads to unstable early training but eventually recovers. Here we ask whether a very small warm start is sufficient by comparing $n_0 = 2$ against the default $n_0 = 8$.

The results are:

Table 4: Effect of prioritized prompt sampling (Eq. (11)) on maj@16 (Qwen3-8B, $D_{half} = 0.12$).

| Sampling scheme | AIME 24 maj@16 | AIME 25 maj@16 |
|---|---|---|
| None | 56.38 | 45.90 |
| Prioritized (SPO) | **67.20** | **55.23** |

Table 5: Comparison with additional RL baselines on AIME 24/25 maj@16 (Qwen3-8B).

| Method | AIME 24 maj@16 | AIME 25 maj@16 |
|---|---|---|
| GRPO (standard) | 55.02 | 44.71 |
| GRPO + Global Norm | 60.59 | 48.32 |
| RLOO | 62.91 | 51.70 |
| SPO (ours) | **67.20** | **55.23** |

- **Early training (step 20):** $n_0 = 8$ reaches avg@16 of $40.42$, whereas $n_0 = 2$ lags behind at $36.46$.
- **Convergence (step 120):** the gap closes and slightly reverses: $47.50$ for $n_0 = 8$ versus $47.71$ for $n_0 = 2$.

Thus, even a minimal warm start ($n_0 = 2$) is sufficient to reach essentially identical final performance, demonstrating that SPO's online tracker is robust. However, a somewhat larger initialization (e.g., $n_0 = 8$) remains preferable in practice, as it stabilizes the initial phase and accelerates early convergence.

## G.4 ADDITIONAL BASELINES: RLOO AND GRPO WITH GLOBAL NORMALIZATION

**RLOO.** Beyond GRPO and $A^*$-PO (Appendix F), we evaluate SPO against RLOO, which uses a leave-one-out group baseline as in Eq. 4 with the same group size $G = 8$. RLOO is a strong group-based baseline but is consistently outperformed by SPO.

**GRPO with batch-level global normalization.** To disentangle the effect of global advantage normalization from the rest of SPO, we augment GRPO with the same batch-level normalization used by SPO: advantages are computed with the standard GRPO group baseline but are then normalized across the whole batch rather than within each group. This variant ("GRPO + Global Norm") substantially improves over standard GRPO, but still falls short of SPO, indicating that global normalization is an important but not exclusive contributor to SPO's gains. Table 5 summarizes the comparison.

These results confirm three points: (i) batch-level normalization alone yields tangible gains even for group-based methods, (ii) single-stream SPO remains competitive against strong variants such as RLOO, and (iii) SPO's remaining advantage primarily stems from its group-free data efficiency and KL-adaptive baseline, which enable $8\times$ more unique prompts per update at fixed rollout budget.

## G.5 EXPERIMENTS ON A SMALLER BASE MODEL: QWEN3-4B

To assess robustness across model scales, we repeat the comparison between GRPO and SPO on Qwen3-4B, using the same Tool-Integrated Reasoning setup and training hyperparameters (including $D_{half} = 0.12$ and the prioritized curriculum). Table 6 reports maj@16 performance on AIME 24 and AIME 25.

Table 6: GRPO vs. SPO on Qwen3-4B (maj@16).

| Method | AIME 24 maj@16 | AIME 25 maj@16 |
|---|---|---|
| GRPO | 56.01 | 46.62 |
| SPO | **59.32** | **48.07** |

SPO again outperforms GRPO on both benchmarks, indicating that the benefits of single-stream optimization and the KL-adaptive value tracker are not specific to the 8B scale. During Qwen3-4B

training we also observe that the model tends to degenerate into a pure-text solver, rarely producing successful tool calls; both methods eventually rely more on text-only reasoning. This suggests that smaller tool-augmented backbones may require a more careful cold-start or auxiliary supervision to reliably acquire tool-use capabilities, which we leave as an avenue for future work.

