# OpenReview forum: "Single-stream Policy Optimization"
_ICLR.cc/2026/Conference — ICLR 2026 Poster_

### Official Review · Reviewer_hydy · 2025-10-29

**Soundness:** 3
**Presentation:** 2
**Contribution:** 2
**Rating:** 6
**Confidence:** 4

**Summary:**

This paper introduces Single-Stream Policy Optimization (SPO), a group-free fine-tuning algorithm that effectively improves the throughput and scalability of group-based methods (e.g., GRPO) in long-horizon or tool-integrated scenarios. The authors also propose a KL-adaptive value tracker and batch-level advantage normalization to provide stable and continuous learning signals. Experimental results show that SPO significantly enhances the model's reasoning accuracy in terms of pass@k.

**Strengths:**

- The idea and algorithm framework is simple.
- The motivation is clear.
- The experimental results are significant.

**Weaknesses:**

- Judging from the title, the core contribution lies in single-stream. Although Section 5.4 discusses the agentic training scenario, I believe plotting runtime versus pass@k is important, as it helps highlight the advantages of single-stream (If the advantage of single-stream mainly lies in distributed training scenarios). The authors mainly analyze a specific example in Figure 4 and Figure 5, but I think providing the corresponding success rate metrics is also necessary.
- I encourage the authors to plot ablation experiments similar to Figure 2, which would help everyone intuitively and quickly understand the contribution of each component in SPO.

**Questions:**

- I think the value tracker and batch-level advantage normalization could also be directly integrated into GRPO, would this lead to further improvements, would it outperforms SPO?
- In a standard RLVR scenario, does single-stream result in $\times G$ times the runtime compared to GRPO (where $G$ is the number of responses in a group)?

---

> ### Author Response · Authors · 2025-11-21
> **[1/2] Rebuttal**
>
> We thank the reviewer for their positive assessment of our framework's simplicity and the significance of our results. We appreciate the opportunity to clarify the runtime advantages in agentic scenarios and to further isolate the benefits of our normalization scheme.
>
> **Q1: Runtime vs. Accuracy and agentic simulations**
>
> We thank the reviewer for this suggestion. We would like to clarify the scope of our experiments and the specific purpose of the simulations.
>
> **Single-Stream Advantages:** First, we wish to emphasize that the advantages of the Single-stream paradigm are foundational and extend beyond just runtime speed. As demonstrated in our main results (Table 1), SPO achieves higher accuracy and data efficiency even in standard settings by providing **lower variance gradients** and **higher data utilization** (avoiding the ~80% computation waste from degenerate groups, as analyzed in Section 5.3). While Section 5.4 highlights SPO’s runtime benefits in agentic settings, our **primary** contributions are algorithmic: the KL-adaptive value tracker and batch-level advantage normalization that improve variance, data efficiency, and final accuracy even in standard RLVR setups.
>
> **Agentic Context & Simulations:** Regarding the "Agentic" analysis, it is important to note that our main experiments (Table 1 and Figure 2) **are already conducted on Tool-Integrated Reasoning (TIR) tasks** where the model interacts with a Python interpreter. Ideally, this would provide the "Runtime vs. Pass@k" data. However, Python interpreter calls are generally fast and low-variance. The critical "synchronization bottleneck" of Group-based methods (GRPO) emerges primarily in **high-variance latency environments**, such as web browsing agents or long-horizon software engineering tasks, where a single slow trajectory stalls the entire group.
>
> While we agree that an end-to-end runtime–accuracy comparison on a live web-agent benchmark would further strengthen the work, running such experiments at the necessary scale is beyond what we can implement within the rebuttal period and our current compute budget. Instead, this submission focuses on (i) **large-scale, single-domain agentic training experiments** on tool-integrated math reasoning, where all main results in Table 1 and Figure 2 already involve interaction with a Python interpreter, and (ii) **structural simulations** in Figures 4 and 5 that isolate the synchronization effect and show a 4.35× potential throughput gain for SPO in high-latency settings. Together with the accuracy gains in Table 1, these results support our claim that SPO offers a better runtime–accuracy trade-off for realistic agentic training, and we view extending this evaluation to full web-based agents as important future work.
>
> **Q2: Ablation Plots**
>
> We appreciate the suggestion to visualize ablations intuitively. We believe the current presentation in **Figure 6 (Appendix F)** serves this purpose effectively by showing the training dynamics (using avg@16) over time.
>
> While Figure 2 uses pass@k to show final performance scaling, we explicitly chose avg@16 / maj@16 curves for the ablation studies because they reveal the **stability and convergence speed** of the learning signal. For instance, Figure 6b demonstrates how removing the baseline leads to immediate volatility. These dynamic training curves provide the most rigorous evidence for *why* each component is necessary, offering insights into stability that a static final accuracy plot might miss.

---

> ### Author Response · Authors · 2025-11-21
> **[2/2] Rebuttal**
>
> **Q3: Integration of batch-level global normalization into GRPO.**
>
> We appreciate this insightful question regarding the modularity of our contributions. To address this, we conducted an experiment integrating **Batch-Level Global Normalization** into GRPO ("GRPO + Global Norm") to see if normalizing advantages across the entire batch rather than within the group closes the performance gap. The results are summarized below:
>
> | **Method** | **AIME24 maj@16** | **AIME25 maj@16** |
> | --- | --- | --- |
> | **GRPO (Standard)** | 55.02 | 44.71 |
> | **GRPO + Global Norm** | 60.59 | 48.32 |
> | **SPO (Ours)** | **67.20** | **55.23** |
>
> Our results confirm that integrating global normalization significantly boosts GRPO's performance (e.g., +5.57 pp on AIME24 maj@16), validating the reviewer's intuition that stable global statistics improve learning. The advantages of Global Advantage Norm have also been mentioned in "Lite PPO" [1]. vHowever, even with this enhancement, "GRPO + Global Norm" significantly underperforms SPO, with SPO maintaining a lead of **+6.6 pp** on AIME24 (maj@16) and **+6.9 pp** on AIME25 (maj@16). This demonstrates that while global normalization is a contributing factor, it does not account for the full performance gain of our method. SPO’s remaining advantage stems from the inherent data efficiency of the single-stream framework, which processes significantly more unique prompts per training step ($8\times$ in this setting) compared to the group-based approach that must repeat prompts to construct a baseline. Adding our full value tracker to GRPO would move the method very close in spirit and implementation to SPO, since the group structure no longer plays a meaningful role in baseline estimation.
>
> [1] Liu, Zihe, et al. "Part i: Tricks or traps? a deep dive into rl for llm reasoning." *arXiv preprint arXiv:2508.08221* (2025).
>
> **Q4: Runtime comparison ($G\times$ slower?).**
>
> **No, SPO does not increase runtime.** The term "Single-stream" refers to the statistical independence of the gradient estimator, not sequential execution. For a fixed total batch size of $B$ samples, GRPO processes $B/G$ prompts with $G$ responses each, whereas SPO processes $B$ unique prompts with 1 response each. Since both methods generate the exact same total number of samples ($B$) in parallel, the computational cost is identical. Furthermore, as analyzed in **Section 4.4** and **Figure 5**, SPO actually improves wall-clock throughput by eliminating the group synchronization barrier, allowing the system to process samples asynchronously without waiting for the slowest generation in a group to finish.

---

> > ### Comment · Reviewer_hydy · 2025-11-25
> >
> > Thank you for your response. I will maintain my score.

---

### Official Review · Reviewer_GDbD · 2025-10-29

**Soundness:** 3
**Presentation:** 3
**Contribution:** 3
**Rating:** 4
**Confidence:** 3

**Summary:**

The paper proposes SPO, a simplified and scalable alternative to group-based methods like GRPO for fine-tuning LLM. It eliminates group synchronization and potential baseline issues by using a KL-adaptive value tracker and global advantage normalization. Experiments with Qwen3-8B show faster convergence and higher accuracy on challenging math reasoning benchmarks, achieving significant gains in maj@32 and pass@k metrics.


Overall, the idea is interesting and potentially offers significant advantages. The acceleration in training speed can be substantial; however, the number of experiments conducted does not seem sufficient. Beyond the limited number of datasets and models, the authors introduced several new hyperparameters but did not isolate their individual contributions. The algorithm is always presented as a whole, while it is limited in how each parameter impacts performance. It would also have been valuable to include an ablation study on a smaller, less computationally expensive experimental setup, as I understand that the current configuration may be time- and resource-intensive.

**Strengths:**

- Training speed is notably increased through the streaming of a single output.
- The paper provides a variance analysis of GRPO.

**Weaknesses:**

- Only one model and one dataset were used for training; testing multiple models across different datasets would be necessary to validate the generality of the results.
- No ablation study was conducted for the D_half parameter.
- The effect of using the same prompt versus different prompts was not analyzed.
- No ablation was performed for the w parameter.
- Since the warm start seems effective, it would be helpful to include an ablation and clarification showing the minimal effective value,  for instance, demonstrating that a relatively small warm start is sufficient to gain its benefits.

**Questions:**

In addition to clarifying the absence of ablation studies, please address the last weakness.

---

> ### Author Response · Authors · 2025-11-21
> **[1/2] Rebuttal**
>
> We thank the reviewer for their thoughtful comments and for recognizing the substantial acceleration in training speed offered by our method. We appreciate the suggestion to conduct more granular ablations to isolate our contributions.
>
> **Q1: Experiments on different base models (Qwen3-4B).**
>
> We agree that testing additional base models strengthens the empirical support for SPO, and we have therefore run a new comparison between SPO and GRPO on Qwen3-4B. On AIME24, GRPO achieves maj@16 of 56.01, whereas SPO reaches 59.32; on AIME25, GRPO obtains 46.62 while SPO attains 48.07. During Qwen3-4B training we also observed that the model almost degenerates into a pure-text solver because it struggles to sample successful tool calls, suggesting that smaller tool-augmented backbones may require cold-start to reliably acquire tool-use capabilities; we have clarified this limitation and the corresponding experimental observation in the revised version.
>
> **Q2: Sensitivity study for $D_\text{half}$.**
>
> We thank the reviewer for this suggestion and have conducted an ablation over $D_\text{half}\in$ {0.06,0.08,0.10,0.12}; the results are relatively stable across this range, with $D_\text{half}=0.12$ giving the best overall performance. All runs in this ablation use the updated prioritized sampling scheme that incorporates $N_\text{eff}$ into the sampling weight.
>
> | **$D_\text{half}$** | **AIME24 maj@16** | **AIME25 maj@16** |
> | --- | --- | --- |
> | **0.06** | 57.58 | 51.28 |
> | **0.08** | 60.64 | 46.24 |
> | **0.10** | 58.90 | 52.73 |
> | **0.12** | **67.20** | **55.23** |
>
> We interpret $D_\text{half}$ as a synchronization parameter that balances estimator variance against bias arising from policy drift. For our specific training configuration in the rebuttal phase(defined by the learning rate and batch size), a value of 0.12 appears to be the optimal "half-life," allowing the tracker to accumulate sufficient historical data to minimize variance without retaining stale estimates from a policy that has already diverged significantly.
>
> **Q3: Effect of "Same Prompt" (GRPO) vs. "Different Prompts" (SPO).**
>
> We appreciate this question as it allows us to clarify how we controlled for computational fairness. Our experimental design ensures that both methods utilize the exact same computational budget per optimization step: GRPO generates 8 responses for each of 256 prompts ($256 \times 8 = 2,048$ total rollouts), while SPO generates 1 response for each of 2,048 unique prompts ($2,048 \times 1 = 2,048$ total rollouts).
>
> The difference in prompt diversity is therefore not an arbitrary experimental setting, but the **primary structural advantage** of the SPO framework. GRPO is mathematically constrained to "spend" its compute budget on redundant sampling (repeating the same prompt 8 times) to construct a low-variance baseline on the fly. In contrast, SPO decouples the baseline estimation from the immediate batch via the value tracker, freeing the algorithm to "spend" that same compute budget on exploring the data distribution more broadly.
>
> **Q4: Ablation for the $w$ parameter.**
>
> We thank the reviewer for this excellent suggestion. To isolate the impact of our automatic curriculum, we conducted an ablation study comparing the full SPO algorithm against a version without sampling mechanism.
> The results confirm the curriculum's critical importance. On the AIME24 benchmark, removing prioritized sampling caused the maj@16 score fall from 67.20 to 56.38. A similar decline was observed on the AIME25 benchmark, where maj@16 fall from 55.23 to 45.90.
> This significant performance degradation across both benchmarks clearly **demonstrates the value of the curriculum**. By focusing training on the most informative prompts, our method improves learning efficiency and helps the model develop more robust, generalizable reasoning skills. We have added this important ablation study to the paper to quantify the benefits of our prioritized sampling strategy. Thank you for helping us strengthen this aspect of our work.

---

> ### Author Response · Authors · 2025-11-21
> **[2/2] Rebuttal**
>
> **Q5: Ablation on warm start sample size ($n_0$) .**
>
> We appreciate the reviewer’s suggestion to investigate the minimal effective size for the warm start. To address this, we conducted an additional ablation study reducing the offline initialization sample size ($n_0$) from our default of 8 down to 2.
>
> As discussed in Appendix F (Figure 6c) of the original submission, we established that removing offline initialization entirely ($n_0=0$) leads to initial training instability and a temporary performance dip, though the model eventually recovers as the online tracker adapts.
>
> To address your question regarding a minimal effective value, we conducted a new ablation experiment using a very small sample size of $n_0=2$ and compared it with our default setting of $n_0=8$. The results on the AIME 25 benchmark (avg@16) are as follows:
>
> - **Early Training** (Step 20)**:** The model with $n_0=8$ significantly outperformed the minimal initialization, achieving 40.42 compared to 36.46 for $n_0=2$. This confirms that a more accurate initial baseline helps stabilize the early learning phase.
> - **Convergence** (Step 120)**:** As training progressed, the performance gap closed completely. The model with $n_0=2$ achieved 47.71, effectively matching the $n_0=8$ model (47.50).
>
> These results demonstrate that while a minimal warm start is sufficient to achieve similar final performance (demonstrating the robustness of SPO’s adaptive tracker), a slightly larger initialization is preferable to avoid early-stage volatility and accelerate the initial convergence.

---

### Official Review · Reviewer_uRF7 · 2025-10-31

**Soundness:** 4
**Presentation:** 4
**Contribution:** 4
**Rating:** 10
**Confidence:** 4

**Summary:**

Recent propular RL methods like GRPO employ group-based value estimate and avoid maintainence and training overhead of a learned value model. However, they also come with some downsides. First, the value calculation in GRPO requires a group of rollouts for each update, which makes it expensive. This becomes especially problematic in agentic workflows when the slowest rollout in the group will slowdown the entire update. Second, there can be degenerate groups (all pass or all fail) which will lead to zero advantage and subsequent wasted computation.

To fix the above limitations, authors propose Single-stream Policy Optimization (SPO). The key innovations of this methods are as follows:
1. It uses a KL-adaptive value estimate as the posterior of the a parametrized beta distribution. The alpha and beta values are updated based on sampled reward (binary 0,1) such that effective update rule for value follows an adaptive exponential moving average.
2. Since each prompt has a unique moving value estimate, they only use single response per prompt and normalize the advantage over the entire training batch. They used PPO's clipped objective for training their policy.
3. They also leverage the value estimate to construct a prompt sampling weight that puts highest weight for prompt close to 50% win rate and creates an automatic prompt curriculum.

Overall, their method consistently improves over GRPO over a diverse range of cometition math evals. In appendix they also compare with A*PO, another method that uses fixed precomputed V* estimate and takes one sample per prompt during online learning. SPO performs very similar to A*PO while also giving the advantages of automatic prompt weighting and learning free dynamic value estimate.

**Strengths:**

- Clear motivation and a very innovative way of calculating a moving model-free value estimate via on-policy samples
- Their normalized advantage calculation leads to low variance and stable training
- Their dynamic value estimate paves way for automatic prompt curriculum

**Weaknesses:**

- Their method does require some value warmstarting to be effective (similar to the inital cost of A*PO).

**Questions:**

1. Would have preferred to see an ablation experiment of w/ and w/o prompt curriculum to see the effectiveness of automatic curriculum
2. I'm curious if the method will very rarely re-visit harder prompts due to the weighted sampling scheme. For example, if a prompt is harder for initial checkpoint and consistently gets 0 reward, its value will drift towards 0 and will be less likely to get resampled in the future.

Typos:
1. (equation 7) $N_{eff} = \alpha(x) + \beta(x)$.. there shouldn't be a +1
2. (equation 10) Advantage shouldn't have subscript $t$ as its only using terminal reward.

---

> ### Author Response · Authors · 2025-11-21
> **[1/1] Rebuttal**
>
> We thank the reviewer for their excellent rating and for highlighting the innovation of our value estimation method and curriculum. We also appreciate the reviewer pointing out the typos in Equation 7 and Equation 10; **we have corrected these errors in the revised PDF.**
>
> **Q1: Ablation of prompt curriculum.**
>
> We thank the reviewer for this excellent suggestion. To isolate the impact of our automatic curriculum, we conducted an ablation study comparing the full SPO algorithm against a version without sampling mechanism.
> The results confirm the curriculum's critical importance. On the AIME24 benchmark, removing prioritized sampling caused the maj@16 score fall from 67.20 to 56.38. A similar decline was observed on the AIME25 benchmark, where maj@16 fall from 55.23 to 45.90.
> This significant performance degradation across both benchmarks clearly **demonstrates the value of the curriculum**. By focusing training on the most informative prompts, our method improves learning efficiency and helps the model develop more robust, generalizable reasoning skills. We have added this important ablation study to the paper to quantify the benefits of our prioritized sampling strategy. Thank you for helping us strengthen this aspect of our work.
>
> **Q2: Revisiting hard prompts.**
>
> We thank the reviewer for the strong support and this keen observation. You are entirely correct that without intervention, the variance-based weighting $\sqrt{\hat{v}(1-\hat{v})}$ would cause consistently failed prompts (where $\hat{v} \approx 0$ ) to vanish from the training loop.
>
> To prevent this, as noted in Equation 11 (Section 4.3), we explicitly add an **exploration bonus** $\epsilon$. This parameter ensures that even prompts with near-zero estimated success retain a fixed minimum probability of being sampled. Consequently, the model is guaranteed to periodically revisit these "hard" prompts, allowing the value tracker to update if the model's capabilities improve.

---

### Official Review · Reviewer_ss5K · 2025-11-01

**Soundness:** 3
**Presentation:** 3
**Contribution:** 3
**Rating:** 4
**Confidence:** 3

**Summary:**

This paper introduces Single-stream policy optimization (SPO) for RLVR (verifiable rewards) LLM training. It basically compares with GRPO,  and replaces GRPO's baseline with (i) a persistent, KL-adaptive Beta tracker for per-prompt success probability (ii) global advantage normalization, and (iii) prioritized prompt sampling that biases training toward prompts with higher learning potential.

**Strengths:**

1. Very clear problem framing. It identifies two GRPO pain points, degenerate groups (no signal) and synchronization overhead and direct addresses both.
2. The group-free simulation (4.35x faster) is compelling for tool-use/agentic regimes as low-latency is needed.

**Weaknesses:**

1. Prioritized sampling is not so valid and might fail. This is a proxy for Bernoulli uncertainty but ignores the effective sample size $N_{eff} = \alpha + \beta$ carried by the Beta tracker. A prompt with $\hat{v} \approx 0.5$ but huge $N_{eff}$ should be less prioritized than one with the same $\hat{v}$ and tiny $N_{eff}$. The prioritized sampling can over-sample well-known, mid-probability prompts.
2. For $\rho(x) = 2^{\frac{-D(x)}{D_{half}}}$, there's no derivation or sensitivity study for $D_{half}$. Can the authors provide more ablation study on that?
3. The experiment is only on one base model (Qwen3-8B) and binary RLVR reward. Showing another base LLM or a non-binary reward variant would be more beneficial.
4. Missing baselines. Can the authors provide some non-GRPO baselines? A*-PO, GRESO, RLOO? Why the authors only provide GRPO as baseline?

**Questions:**

See above weaknesses.

---

> ### Author Response · Authors · 2025-11-21
> **[1/2] Rebuttal**
>
> We thank the reviewer for their detailed feedback, particularly for the theoretical insights regarding our sampling strategy and for suggesting additional baselines to strengthen our empirical evaluation.
>
> **Q1: Incorporation of effective sample size ($N_\text{eff}$)  in prioritized sampling.**
>
> We thank the reviewer for highlighting that our prioritized sampling heuristic only uses the Bernoulli variance proxy and does not explicitly incorporate the effective sample size $N_\text{eff}=\alpha+\beta$ from the Beta tracker. We agree that **it is reasonable to down-weight prompts that are already well-estimated** even when $\hat{v}\approx 0.5$.
>
> To address the reviewer’s concern, we additionally implemented
>
> $$
> w_i(x)\propto \frac{\sqrt{\hat{v}\_{-1}(x)(1-\hat{v}\_{-1}(x))}}{N_\text{eff}}+\epsilon.
> $$
>
> On AIME24, maj@16 increase from 55.95 (without  $N_\text{eff}$ ) to 60.64 (with $N_\text{eff}$) ; on AIME25, they drop  from 52.75 to 46.24. $D_\text{half}$ is set to 0.08 here.
>
> These mixed results highlight a nuanced **exploration-exploitation trade-off**. Incorporating $N_\text{eff}$ encourages the model to cycle through prompts more quickly (Exploration), preventing it from spending too many steps on any single problem once the value is estimated. This broader coverage appears beneficial for **AIME 24**, a trend that aligns with findings in MathArena [1] which suggest a stronger correlation between the DAPO training set and AIME 24.
>
> We agree that explicitly encoding estimation confidence via $N_\text{eff}$ is a **principled refinement** of our heuristic, we have modified the sampling weight in the script to incorporate $N_\text{eff}$ in a manner that balances this need for exploration with the deep learning required for generalization.
>
> [1] Balunović, Mislav, et al. "Matharena: Evaluating llms on uncontaminated math competitions." *arXiv preprint arXiv:2505.23281* (2025).
>
> **Q2: Sensitivity study for $D_\text{half}$.**
>
> We thank the reviewer for this suggestion and have conducted an ablation over $D_\text{half}\in$ {0.06,0.08,0.10,0.12}; the results are relatively stable across this range, with $D_\text{half}=0.12$ giving the best overall performance. All runs in this ablation use the updated prioritized sampling scheme that incorporates $N_\text{eff}$ into the sampling weight.
>
> | **$D_\text{half}$** | **AIME24 maj@16** | **AIME25 maj@16** |
> | --- | --- | --- |
> | **0.06** | 57.58 | 51.28 |
> | **0.08** | 60.64 | 46.24 |
> | **0.10** | 58.90 | 52.73 |
> | **0.12** | **67.20** | **55.23** |
>
> We interpret $D_\text{half}$ as a synchronization parameter that **balances estimator variance against bias arising from policy drift**. For our specific training configuration in the rebuttal phase(defined by the learning rate and batch size), a value of 0.12 appears to be the optimal "half-life," allowing the tracker to accumulate sufficient historical data to minimize variance without retaining stale estimates from a policy that has already diverged significantly.
>
> **Q3: Experiments on different base models (Qwen3-4B).**
>
> We agree that testing additional base models strengthens the empirical support for SPO, and we have therefore run a new comparison between SPO and GRPO on Qwen3-4B. On AIME24, GRPO achieves maj@16 of 56.01, whereas SPO reaches 59.32; on AIME25, GRPO obtains 46.62 while SPO attains 48.07. During Qwen3-4B training we also observed that the model almost degenerates into a pure-text solver because it struggles to sample successful tool calls, suggesting that smaller tool-augmented backbones may require cold-start to reliably acquire tool-use capabilities; we have clarified this limitation and the corresponding experimental observation in the revised version.
>
> While we did not have compute to run a full non-binary RLVR experiment during rebuttal, Section 4.1 already shows how the tracker reduces to a general EMA when rewards are real-valued. We agree that non-binary experiments would be valuable and intend to run them in future work.

---

> ### Author Response · Authors · 2025-11-21
> **[2/2] Rebuttal**
>
> **Q4: Missing non-GRPO baselines (RLOO, $A^*$-PO).**
>
> We appreciate the request for non-GRPO baselines. In addition to focusing on GRPO in the main text because it is the most widely adopted group-based RLVR method, we have already included ablations with *A**-PO in Appendix F, where SPO provides a more stable and
> effective learning signal in environments that demand significant policy evolution. Reviewer uRF7 also mentions this in the review.
>
> We have now also compared SPO with RLOO under the same setup. On AIME24, RLOO achieves maj@16 of 62.91, while SPO attains 67.20; on AIME25, RLOO reaches 51.70, whereas SPO reaches 55.23. We will add these results and clarify in the revision that SPO is competitive with, and often outperforms, strong non-GRPO baselines such as *A**-PO and RLOO.
>
> GRESO focuses on **filtering uninformative prompts before generation**, which is complementary to our baseline design; integrating GRESO on top of SPO would be an interesting direction, but is beyond our current compute budget.

---

> > ### Comment · Reviewer_ss5K · 2025-11-23
> > **Thanks for the authors responses.**
> >
> > Thanks for the authors responses! I still insist on seeing experimetal results on adding more non-binary rewards and other base models other than Qwen3-8B/ Qwen3-4B. I believe there are other model families for solving math tasks.

---

> > > ### Author Response · Authors · 2025-11-28
> > > **Thanks for the reviewer's comment.**
> > >
> > > **Q5: Non-binary reward**
> > >
> > > We thank the reviewer for requesting experiments beyond pure binary rewards. In addition to our main binary-reward setup, we have conducted new experiments with a non-binary reward that follows the official ReTool [1] implementation (as reproduced in `verl`) to encourage tool usage.
> > >
> > > Concretely, we define the shaped reward as where $s \in $\{1,-1\} is the correctness signal (correct = 1, wrong = -1), and $T$ is the number of tool calls. In this setting the reward is real-valued, so we instantiate SPO’s value tracker using the EMA formulation in Section 4.1 (Eq. (7)) to directly track $\hat v$ for non-binary rewards.
> > >
> > > $$
> > > r_{\text{non-binary}}
> > > = \frac{\min\Bigl(-0.6,\; s + \frac{T - 2}{2} \times 0.1\Bigr) + 1}{2},
> > > $$
> > >
> > > Under this non-binary reward, we observe that SPO continues to outperform GRPO:
> > >
> > > - On AIME24, GRPO achieves maj@16 of 78.48, while SPO reaches 83.76.
> > > - On AIME25, GRPO achieves maj@16 of 69.55, while SPO achieves 74.39.
> > >
> > > Thus, even when moving from binary to shaped, non-binary rewards that explicitly incentivize tool calls, SPO remains consistently better than GRPO. These results empirically support our claim that SPO is robust to the reward specification and continues to provide a stronger learning signal in both binary and non-binary reward settings.
> > >
> > > **Q6: Experiment with Llama-3.1-8B on simpleRL dataset.**
> > >
> > > We thank the reviewer for encouraging us to evaluate SPO on additional base model families beyond Qwen3-8B/4B. To this end, we conducted an additional experiment starting from the **Llama-3.1-8B** base model on the **simpleRL** benchmark [2]. On this setup, **GRPO** attains **27.02 pass@1**, whereas **SPO** reaches **28.02 pass@1** on the simpleRL test set, giving a +1.0 percentage point absolute improvement. It is fully consistent with our overall findings: SPO continues to provide a stronger and more stable learning signal than GRPO even when (i) switching to a different model family (Llama vs. Qwen) and (ii) moving to a distinct reasoning-oriented dataset.
> > >
> > >
> > > [1] Feng, Jiazhan, et al. "ReTool: Reinforcement Learning for Strategic Tool Use in LLMs" *arXiv preprint arXiv:2504.11536* (2025).
> > >
> > > [2] Zeng, Weihao, et al. "SimpleRL-Zoo: Investigating and Taming Zero Reinforcement Learning for Open Base Models in the Wild" *arXiv preprint arXiv:2503.18892* (2025).

---

### Author Response · Authors · 2025-11-21
**Overall Response to All Reviewers**

We thank all reviewers for their constructive feedback and encouraging assessment of our work. We are particularly grateful for the recognition of our method's innovation and efficiency:

- **Reviewer ss5K** commended the "**very clear problem framing**" that "**directly addresses**" the critical flaws of GRPO, and found our group-free simulation "**compelling for tool-use/agentic regimes**."
- **Reviewer uRF7** highlighted the "**clear motivation**" and our "**very innovative way of calculating a moving model-free value estimate**," noting that our method "**consistently improves over GRPO**" and that the dynamic value estimate "**paves way for automatic prompt curriculum**."
- **Reviewer GDbD** acknowledged that the idea offers "**potentially significant advantages**," and that the paper provides a valuable "**variance analysis of GRPO**."
- **Reviewer hydy** noted that "**the idea and algorithm framework is simple**," "**the motivation is clear**," and "**the experimental results are significant**," confirming that SPO effectively improves scalability in long-horizon scenarios.

We have carefully addressed the questions raised by the reviewers and **have integrated the following supplementary experiments and revisions into the updated PDF (highlighted by blue color)**:

1. **New Model Scale:** Results on **Qwen3-4B** to demonstrate generalization across model sizes.
2. **New Baselines:** Comparisons against **RLOO**, and an enhanced "**GRPO + Global Norm**" baseline.
3. **Detailed Ablations:** Analyses of the **curriculum design** ($N_{\text{eff}}$ and $w$), **hyperparameter sensitivity** ($D_{\text{half}}$), and **initialization size** ($n_0$).

For the additional experiments conducted during the rebuttal phase, we slightly reduced the training scale due to time constraints. For GRPO (with and without global normalization) and RLOO, we use a total batch size of 128 with a mini-batch size of 16 and a group size of 8. For SPO, we set the batch size to 1024 and the mini-batch size to 128 so that the total number of sampled trajectories per update matches GRPO (128 prompts × 8 samples). In all settings, the maximum response length is set to 8,192 tokens.

---

### Author Response · Authors · 2025-12-01
**Summary of Rebuttal and Revision**

We thank the reviewers for their constructive feedback and high engagement, which has significantly strengthened our submission. In response to their comments, we have uploaded a revised manuscript. Below, we summarize the key updates and additional experiments conducted during the discussion phase.

**1. Broader Generalization: New Models and Reward Functions**

To demonstrate that SPO’s benefits extend beyond specific setups, we expanded our evaluation as requested by Reviewer **ss5K** and **GDbD**:

- **New Base Models and Benchmark:** We evaluated SPO on **Qwen3-4B** and **Llama-3.1-8B**. SPO consistently outperforms GRPO across these different model families and sizes. Notably, on Llama-3.1-8B (simpleRL benchmark), SPO achieves a +1.0 pass@1 improvement over GRPO.
- **Non-Binary Rewards:** We implemented a shaped, real-valued reward function (following ReTool) to penalize incorrect tool usage. SPO successfully utilizes the EMA-based value tracker to outperform GRPO significantly in this non-binary setting (+5.28 maj@16 on AIME24).

**2. Stronger Baselines**
We addressed concerns regarding baseline coverage (Reviewers **ss5K**, **hydy**):

- **RLOO:** We added a comparison with RLOO. SPO outperforms RLOO on both AIME24 (+4.29 maj@16) and AIME25 (+3.53 maj@16).
- **GRPO + Global Normalization:** We integrated batch-level advantage normalization into GRPO. While this improves GRPO, SPO retains a significant lead (~6.6 points), confirming that our gains stem from the data efficiency of the single-stream framework (processing $8\times$ more unique prompts) rather than normalization alone.

**3. Refined Methodology and Deepened Analysis**

- **Improved Sampling Strategy:** Incorporating theoretical insights from Reviewer **ss5K**, we refined our prioritized sampling heuristic to account for the effective sample size ($N_{\text{eff}}$), balancing exploration with exploitation.
- **Curriculum Ablation:** As requested by Reviewers **uRF7** and **GDbD**, we isolated the impact of our prompt curriculum. Removing prioritized sampling caused a drastic performance drop (~10 points), empirically validating the curriculum as a critical component.
- **Hyperparameter Sensitivity:** We conducted sensitivity analyses for the half-life parameter $D_{\text{half}}$ and the warm-start size $n_0$, showing the method is robust within reasonable ranges.

**4. Clarifications**
We clarified experimental controls regarding computational fairness (Reviewer **GDbD**), confirming that SPO and GRPO use identical total rollout budgets. We also clarified runtime advantages in agentic settings and corrected typos (Reviewer **uRF7**).

---

### Meta-Review · Area_Chair_kfyr · 2026-01-07

**Summary:**

The reviewers raise concern on limited evaluation due to limited numbers of datasets and models, and the lack of ablation of hyperparameters. The rebuttal sufficiently addressed these concerns with extra experiments. I think in particular the inclusion of additional models help providing evidence substantially. I recommend acceptance based on the these revisions.

**Reviewer Concerns:**

See summary.

**Reviewer Scores:**

Reviewer ss5K would likely increase their score. The concerns there are concrete and they've been addressed by rebuttal.
The rest will retain their original positive evaluation.

---

### Decision · Program_Chairs · 2026-01-26

Accept (Poster)